# Polymorphisms in Processing and Antigen Presentation-Related Genes and Their Association with Host Susceptibility to Influenza A/H1N1 2009 Pandemic in a Mexican Mestizo Population

**DOI:** 10.3390/v12111224

**Published:** 2020-10-29

**Authors:** Marco Antonio Ponce-Gallegos, Aseneth Ruiz-Celis, Enrique Ambrocio-Ortiz, Gloria Pérez-Rubio, Alejandra Ramírez-Venegas, Nora E. Bautista-Félix, Ramcés Falfán-Valencia

**Affiliations:** 1HLA Laboratory, Instituto Nacional de Enfermedades Respiratorias Ismael Cosio Villegas, Mexico City 14080, Mexico; marcoapg@iner.gob.mx (M.A.P.-G.); aseneth_rc@hotmail.com (A.R.-C.); e_ambrocio_iner@hotmail.com (E.A.-O.); glofos@yahoo.com.mx (G.P.-R.); 2Tobacco Smoking and COPD Research Department, Instituto Nacional de Enfermedades Respiratorias Ismael Cosio Villegas, Mexico City 14080, Mexico; aleravas@hotmail.com (A.R.-V.); dra.norabautista@gmail.com (N.E.B.-F.)

**Keywords:** influenza A/H1N1 pdm09, *TAP2*, *TAPBP*, *PSMB8*, host susceptibility

## Abstract

(1) Background: The influenza A/H1N1 pdm09 virus rapidly spread throughout the world. Despite the inflammatory and virus-degradation pathways described in the pathogenesis of influenza A virus (IAV) infection, little is known about the role of the single nucleotide polymorphisms (SNPs) in the genes involved in the processing and antigenic presentation-related mechanisms. (2) Methods: In this case-control study, we evaluated 17 SNPs in five genes (*TAP1*, *TAP2*, *TAPBP*, *PSMB8*, and *PSMB9*). One hundred and twenty-eight patients with influenza A/H1N1 infection (INF-P) and 111 healthy contacts (HC) were included; all of them are Mexican mestizo. (3) Results: In allele and genotype comparison, the rs241433/C allele (*TAP2*), as well as AG haplotype (rs3763365 and rs4148882), are associated with reduced risk for influenza A/H1N1 infection (*p* < 0.05). On the other hand, the rs2071888G allele (*TAPBP*) and GG haplotype (rs3763365 and rs9276810) are associated with a higher risk for influenza A/H1N1 infection. In addition, after adjustment for covariates, the association to a reduced risk for influenza A/H1N1 infection remains with rs241433/C allele (*p* < 0.0001, OR = 0.24, 95% CI = 0.13–0.43), and the association with *TAPBP* is also maintained with the G allele (*p* = 0.0095, OR = 1.89, 95% CI = 1.17–3.06) and GG genotype models (*p* < 0.05, OR = 2.18, 95% CI = 1.27–3.74). (4) Conclusion: The rs241433/C allele and AC genotype (*TAP2*) and the AG haplotype are associated with a reduced risk for influenza A/H1N1 infection. In addition, the rs2071888/G allele and GG genotype (*TAPBP*) and the GG haplotype are associated with a higher risk for developing influenza A/H1N1 infection in a Mexican mestizo population.

## 1. Introduction

In April 2009, the Mexican Secretary of Health reported the emergence of the influenza A/H1N1 pdm09 virus, which shares molecular characteristics of swine, avian, and human influenza viruses, and rapidly spread throughout the world [1,2]. In Mexico, the lethality rate was estimated to be 5% for confirmed cases of influenza A/H1N1, with several comorbidities such as immunosuppression, pre-existing respiratory and cardiovascular diseases, diabetes, tobacco smoking, and obesity, some of the most important predictors of the disease severity [2,3,4]. The most frequent signs and symptoms are fever, dyspnea, cough, myalgias, arthralgias, headache, and dyspnea. The most frequent adverse outcomes are mechanical ventilation, intensive unit care (ICU) admission, and death [1,5].

Additionally, previous studies have described several clinical and laboratory predictors of mortality or disease outcome. These parameters include sex (male), delayed medical attention and treatment onset, intensive care unit admission, low lymphocyte and platelet count, high creatine phosphokinase (CPK), serum creatine, blood urea nitrogen (BUN), and lactate dehydrogenase (LDH) [6,7,8].

Previous studies have determined that one of the most critical pathological mechanisms in this viral infection is systemic dysregulation of the inflammatory response correlated with disease severity [9,10,11]. In this sense, genetic variants in proinflammatory genes have been associated with host susceptibility for influenza A/H1N1 infection. For example, single nucleotide polymorphisms (SNPs) in *TNF*, *LTA*, *IL1B*, and *IL6* are associated with a higher risk for influenza A/H1N1 infection in the Mexican population [11,12].

However, antigen presentation by influenza-infected cells through Human leukocyte antigens (HLA) class I is necessary to mount an effective antiviral immune response. It has been described that cytotoxic T lymphocytes (CTL) kill influenza-infected cells upon recognition of diverse HLA class I peptide complexes at the cell surface, including short viral peptides of 8–12 aa in length [13,14]. The assembly of these peptides occurs in the endoplasmic reticulum (ER) of infected cells and involves the transporter associated with antigen processing (TAP) and Tapasin (encoded by *TAPBP* gene), as well as ER chaperones. TAP is a critical factor comprising two subunits, TAP1 and TAP2 (encoded by *TAP1* and *TAP2* genes, respectively), required for translocation of peptides from the cytosol into the ER. On the other hand, Tapasin is another critical cofactor required for the assembly of HLA class I with exogenous peptides, which were obtained by intracellular degradation through proteasome, formed by proteasome subunit beta type 8 (PSMB8, encoded by *PSMB8* gene) and 9 (PSMB9, encoded by *PSMB9* gene) (previously known as large multifunctional peptidase 7 (LMP7) and 2 (LMP2), respectively) subunits [15,16,17,18].

Various studies have demonstrated that different HLA class I alleles are associated with a higher risk for influenza A/H1N1 infection in several populations [19,20]. Moreover, the role of SNPs in processing and antigen presentation-related genes has been studied in hypersensitivity pneumonitis (HP) [21], ankylosing spondylitis [22], and human papillomavirus (HPV) infection [23,24], but not in influenza A/H1N1 infection.

For all mentioned above, the study aimed to evaluate the association between processing and antigen presentation-related gene (*TAP1*, *TAP2*, *TAPBP*, *PSMB8,* and *PSMB9*) SNPs with the genetic susceptibility for influenza A/H1N1 infection in a Mexican mestizo population.

## 2. Materials and Methods

A retrospective case-control study was performed between May 2009 and March 2012 using nasopharyngeal swab samples from patients hospitalized with clinical symptoms of influenza and confirmed diagnosis by PCR techniques as previously stated [19].

### 2.1. Ethics Statements

This study was reviewed and approved by the Institutional Committees for Investigation, Ethics in Research, and Biosecurity of the Instituto Nacional de Enfermedades Respiratorias Ismael Cosío Villegas (INER) (approbation number: B05–10, approved on 2 March 2010). All participants were informed about the protocol’s aim after being given a detailed description of the study and invited to participate as volunteers. All of them signed an informed consent paper and supplied a privacy statement describing the legal protection of personal data; both documents approved by the institutional Research and Ethics in Research Committees.

A trained and responsible physician informed the patient about a possible diagnosis of influenza. All participants were invited to participate voluntarily by a trained researcher. All participants received medical care even if they chose not to participate in the research protocol. A clinical investigation was conducted according to the principles expressed in the Declaration of Helsinki. All experiments were performed following the relevant guidelines and regulations. The STREGA (STrengthening the REporting of Genetic Association) guidelines were considered to design this genetic association study [25].

### 2.2. Diagnosis

All patients underwent the QuickVue Influenza A + B test (Quidel, San Diego, CA, USA). Additionally, peripheral blood from patients and asymptomatic healthy contacts (HC) was collected; these interventions adhered to the guidelines and recommendations of the Centers for Disease Control and Prevention of the United States (CDC) and the World Health Organization (WHO). Subsequently, the RespiFinder assay was done, and the presence of the influenza A/H1N1 virus was confirmed.

### 2.3. Population

Our study’s subjects were evaluated according to a clinical questionnaire, which contained questions regarding influenza vaccination. The inclusion period was from May 2009 to March 2012, so the subjects examined during the study belonged to three different influenza waves in Mexico. A total of 261 subjects with clinical symptoms and signs of influenza were enrolled, of which 128 were positive (INF-P) and 133 were negative for influenza A/H1N1 infection. On admission to the hospital, all patients with suspected influenza were treated with oseltamivir. In this study, most patients sought medical attention 3–4 days after the onset of symptoms. All clinical data were collected retrospectively; nonetheless, blood samples and nasopharyngeal swabs were collected at the hospital during the first 24 h after admission. We also included the variable “died” during the patients’ hospitalization to perform a mortality analysis.

In the healthy contacts (HC) group, 111 biologically unrelated, intrahousehold contacts were enrolled who were asymptomatic and in close, continuous contact with influenza A/H1N1 patients before and during hospital admission. None of the HCs developed symptoms or was hospitalized for influenza virus infection. Peripheral blood from these individuals was tested for anti-A/H1N1 antibody titers by hemagglutination inhibition (HI) assay according to the method described by Julkunen I, 1985 [18] to determine exposure to the virus. All subjects included in this group had HI titers greater than 1:16, indicating direct contact with the virus. According to the samples’ availability in the biobank generated since 2009 in the HLA Laboratory, the participants were selected.

### 2.4. DNA Extraction

Firstly, we obtained 15 mL of peripheral blood via venipuncture in two EDTA tubes (as anticoagulant) from all 472 subjects. After that, the DNA extraction was performed using the commercial BDtract Genomic DNA isolation kit (Maxim Biotech, San Francisco, CA, USA). The DNA was quantified by UV absorption spectrophotometry at the 260-nm wavelength employing a NanoDrop 2000 device (Thermo Scientific, Wilmington, DE, USA). Contamination with organic compounds and proteins was determined by measuring the ratio absorbance at 260/280. Samples were considered of good quality when the ratio was ~1.8. All samples were adjusted to 15 µg/µL for subsequent genotyping.

### 2.5. SNP Selection

Four SNPs (rs1057149, rs2127679, rs4148882, and 41561219) in *TAP1*; four (rs13501, rs241433, rs241441, and rs2071544) in *TAP2*; two (rs2071888 and rs2282851) in *TAPBP*; four (rs2071542, rs2071543, rs3763365, and rs9276810) in *PSMB8;* and three (rs17587, rs241418, and rs2071534) in *PSMB9* genes were selected based on a bibliographic search in PubMed (NCBI), identifying polymorphisms previously associated with other virus infectious and respiratory diseases. The molecular characteristics of the SNPs evaluated are described in Appendix A (frequencies were obtained from the 1000 genomes project [26]).

### 2.6. SNP Genotyping

The allele discrimination was performed using commercial TaqMan probes (Applied Biosystems, San Francisco CA, USA), employing qPCR in a 7300 Real-Time PCR System (Applied Biosystems/Thermo Fisher Scientific Inc., Singapore), and the analysis performed by sequence detection software version 1.4 (Applied Biosystems, San Francisco, CA, USA). Four controls without template (contamination controls) were included for each genotyping plate, and 5% of the genotyped in duplicate as controls for allele assignment.

### 2.7. Bioinformatic Analysis and in Silico Analysis

HaploReg v4.1 [27] was consulted to determine the associated SNPs’ putative biologic effect. In addition, with the RNAfold web server [26], we introduce the section’s sequence that includes the SNPs to elucidate for changes in the secondary RNA structures. We also conducted an in silico analysis with the associated SNPs to evaluate the possible function or biological consequences across diverse software, considering if they are coding or noncoding SNPs. Appendix A shows the algorithm used for the in silico analysis of the associated SNPs.

### 2.8. Hardy–Weinberg Equilibrium and Haplotypes

The Hardy–Weinberg Equilibrium (HWE) was assessed using SNPStats online software for the 17 SNPs evaluated [28]. The haplotype analysis was performed using Haploview software v. 4.2 [29], using the Gabriel et al. [30] established criteria.

### 2.9. Statistical Analysis

Clinical quantitative variables were analyzed using SPSS for Mac Os, v20.0 (SPSS software, IBM, Chicago, IL, USA). Kolmogorov–Smirnov normality test was used, and parametric or nonparametric tests were used as appropriate. Genotype analysis and genetic association models were carried out with Pearson’s chi-squared and Fisher’s exact tests using Epi Info v. 7.1 [31] and Epi Dat v.3.1 software, and 2 × N and 2 × 2 contingency tables were made to estimate the genetic association for Influenza A/H1N1 infection and mortality. A *p*-value < 0.05 was considered statistically significant. Bonferroni correction (multiple testing) was applied to adjust the significant values of the genetic association results. Comparisons were made between case and control groups. After that, a logistic regression analysis was performed to adjust for potential confounding variables (sex and body mass index (BMI)) using Plink v. 1.07 [32].

## 3. Results

### 3.1. Demographic Variables in Case and Control Groups

One hundred and twenty-eight patients with influenza A/H1N1 infection (INF-P) and 111 HCs were included. When we compared demographic data between groups, we found that there were more males in the INF-P group, and they had a higher body mass index (BMI) compared with HCs (*p* < 0.05) (Table 1). Additionally, we found that the most common comorbidity was hypertension (13.28%), followed by asthma, type 2 diabetes mellitus, and COPD. Besides, the most frequent symptom was fever (75.78%), followed by dyspnea (72.66%) and cough (53.13%). Finally, pneumonia was the most frequent complication (59.38%). The noncomparable clinical features of the INF-P group are shown in Appendix A.

### 3.2. Demographic and Clinical Variables in Survivors and Nonsurvivors

Additionally, we performed an intracase analysis to evaluate which factors are associated with a higher risk of mortality in those patients with influenza A/H1N1 virus infection. Interestingly, we found that nonsurvivors subjects presented a lower platelet count, and higher glucose levels, CPK, and LDH than survivors subjects (*p* < 0.05). Most of the nonsurvivors subjects were in ICU, compared with survivors group subjects (*p* < 0.05). Complete results are shown in Appendix B, Table A1.

### 3.3. Hardy–Weinberg Equilibrium

Six polymorphisms evaluated did not meet the HWE (four in *TAP2*—rs13501, rs241433, rs241441, rs2071544; and two in *TAPBP*—rs2071888 and rs2282851) for the HC group. However, allele and genotype comparisons were carried out in all SNPs selected for the study. The results are shown in Appendix A.

### 3.4. Allele and Genotype Frequencies

We compared allele and genotype frequencies between INF-P and HC groups for the 17 SNPs selected for the study. In Table 2, the statistically significant results are shown; whereas in Appendix A, the allele and genotype frequencies for nonsignificative SNPs are shown. For *TAP1*, *PSMB8*, and *PSMB9* SNPs, we did not find statistically significant differences in allele or genotype frequencies between INF-P and HC groups. However, the *TAP2* rs241433/AC genotype and C allele are associated with a reduced risk for influenza A/H1N1 infection (*p* < 0.001, OR = 0.24, 95% CI = 0.14–0.22; OR = 0.44, 95% CI = 0.29–0.66, respectively), and the association is maintained after Bonferroni correction (*p* < 0.001).

On the other hand, *TAPBP* rs2071888/GG genotype and G allele are associated with higher risk for influenza A/H1N1 infection (*p* < 0.001, OR = 5.64, 95% CI = 2.00–15.86; *p* = 0.009, OR = 1.62, 95% CI = 1.13–2.34), and only the genotype association is maintained after Bonferroni correction (*p* = 0.014). In addition, the *TAPBP* rs2282851/T allele is associated with reduced risk for Influenza A/H1N1 infection (*p* = 0.048, OR = 0.69, 95% CI = 0.40–0.99). However, the association is not maintained after Bonferroni correction (*p* > 0.05).

### 3.5. Allele and Genotype Frequencies from Survivors and Nonsurvivors

We compared allele and genotype frequencies of the associated SNPs between Survivors and Nonsurvivors from the INF-P group. We did not find statistically significant differences in allele or genotype frequencies between groups. Complete results are shown in Appendix B, Table A2.

### 3.6. Logistic Regression Analysis

A logistic regression analysis adjusting by possible confounding covariates (sex and BMI) was carried out, which results in significant differences between INF-P and HC groups (Table 3). Interestingly, we found that the association with a reduced risk for Influenza A/H1N1 infection remains significant with the rs241433/C allele (*p* < 0.0001, OR = 0.24, 95% CI = 0.13–0.43). However, genotype analysis was not possible to perform with this SNP because we did not find minor allele homozygous in our population.

On the other hand, association to higher risk for Influenza A H1N1 infection with rs2071888/G allele (*p* = 0.0095, OR = 1.89, 95% CI = 1.17–3.06) and genotype (*p* = 0.005, OR = 2.18, 95% CI = 1.27–3.74) are also maintained.

### 3.7. Logistic Regression Analysis in Survivors and Nonsurvivors

After allele and genotype frequencies comparison between survivors and nonsurvivors, we performed a logistic regression analysis to adjust for possible confounding variables. However, we did not find statistically significant differences between groups (*p* > 0.05, table not shown).

### 3.8. Bioinformatic Analysis

We found that the rs241433/A allele has reduced affinity with the myeloid and B-cell development regulatory unwinding protein 1 (UP1). On the other hand, the rs2071888/C, affinity with hepatocyte nuclear factor 4 (HNF4) protein, a nuclear receptor, and transcription factor are increased (Table 4).

Moreover, no differences in predicted secondary messenger ribonucleotide acid (mRNA) structures were found in rs241433. However, for rs2071888, we can observe differences in the number of loops with rs2071888/C allele (Appendix C, Figure A1D).

### 3.9. In Silico Analysis

The rs2071888 (C/G) change from medium size and polar amino acid (threonine) change to large size, and basis (arginine) could alter the polarity of the protein structure (https://web.expasy.org/variant_pages/VAR_010253.html, https://www.uniprot.org/uniprot/O15533). The arginine presence increases protein stability (Gibbs free energy in 260 position = 0.31 kcal/mol at 25 °C and pH 7) (https://folding.biofold.org/cgi-bin/i-mutant2.0.cgi).

In silico analysis for rs241433 did not show sites of alternative splicing (http://wangcomputing.com/assp/, http://www.cbs.dtu.dk/services/NetGene2/) or micro-RNA (miRNAs) generation (http://www.mirbase.org/, https://tanuki.ibisc.univ-evry.fr/evryrna/mirnafold/mirnafold_form).

### 3.10. Linkage Disequilibrium (LD) and Haplotype Analysis

The haplotype analysis was carried out to determine its association with influenza A/H1N1 infection susceptibility and the LD between the same gene’s SNPs. This analysis included the 11 SNPs evaluated that met the HWE, located in *TAP1*, *PSMB8,* and *PSMB9* genes, comparing INF-P versus HC groups.

Haplotypes and their frequencies are summarized in Figure 1. Figure 1A shows that the haplotype shaped by (both in *PSMB8* gene) is in high LD (r^2^ = 61). Moreover, according to the frequencies of this haplotype in both groups, we found a significative difference between INF-P vs. HC with GG haplotype (conformed by the common allele of two SNPs, *p* = 0.038, OR = 1.95, 95% CI = 1.04–3.64).

On the other hand, the haplotype shaped by rs3763365 and rs4148882 is not in high LD (r^2^ = 49). However, when we compared frequencies of this haplotype in both groups, we found a reduced risk for influenza A/H1N1 infection with AG haplotype (minor alleles of both SNPs, *p* = 0.046, OR = 0.67, 95% CI = 0.45–0.99).

## 4. Discussion

Clinical features of influenza A/H1N1 pdm09 infection associated with higher risk (hospitalization, disease severity, and mortality) have been widely investigated. Our study found that patients with influenza A/H1N1 infection were predominantly smoker men with higher BMI. These findings agree with previous reports, where they are added to the mentioned risk factors—pre-existing pulmonary diseases, diabetes, cardiovascular diseases, immunosuppression, and pregnancy are included [3,4,33].

Additionally, previous studies in the Mexican population have reported that intensive care unit admission; low lymphocyte and platelet count; and high CPK, BUN, and LDH are associated with a higher risk for influenza A/H1N1 virus infection mortality [6,7,8], coinciding with our results.

The HWE was assessed for all the SNPs evaluated. Unfortunately, six of the 17 selected SNPs (in *TAP2* and *TAPBP*) did not meet it (*p* > 0.05). However, we decided to continue with the subsequent analyses due to the SNPs position at the gene structure and their possible biological implications. After allele and genotype frequencies and adjustment results by covariates (sex and BMI), we found that rs2071888/GG genotype in *TAPBP* is associated with increased risk for influenza A/H1N1 infection. Conversely, rs241433/AG genotype in *TAP2* is associated with reduced risk for influenza A/H1N1 infection. Nevertheless, both SNPs did not meet the HWE. Although this could seem a limitation, some authors have described that mestizo populations (such as Mexican) have a rich genetic variability, a product of the years of genetic recombination between ancestral populations (Amerindian, Caucasian, African, and Asian descendants), which could result in some SNPs having a different behavior from those described in the scientific literature [34,35].

TAP complex (shaped by TAP1 and TAP2 subunits) is a necessary transporter for translocating peptides, which were obtained via proteasome degradation, from the cytosol to ER, where HLA class I assembly takes place. Previous studies have demonstrated that in mice and humans, the TAP complex can play a crucial role in the translocation of viral epitopes to ER [36,37], and mutations in *TAP1* or *TAP2* result in structural changes that can alter antigen recognition and presentation. In this sense, SNPs in *TAP2* have been previously associated with a higher risk for ankylosing spondylitis [22] and cervical carcinoma associated with HPV [24]. It has been proposed that the genetic variation in this gene may cause functional alterations, possibly by affecting the ATP-binding capacity and transport efficiency of TAP, as well as the expression and structural stability of the mRNA and protein [38]. The rs241433 has previously only been associated with a higher risk for rheumatoid arthritis [39]. Our work is the first in which this SNP is associated with reduced risk for influenza A/H1N1 infection, and more functional and genetic studies are required to establish its role in the pathogenesis of this infectious disease. In the bioinformatic analysis, the rs241433/A allele has reduced affinity with the myeloid and B-cell development regulatory protein UP1; and on the other hand, the C allele has a high affinity for this protein. Additionally, no differences in predicted secondary mRNA structures were found with the rs241433. The UP1 protein (also known as heterogeneous nuclear ribonucleoprotein A1 (*HNRNPA1*)) plays a crucial role in the regulation of alternative splicing, and multiple alternatively spliced transcript variants have been found [40]. However, there are no previous studies in literature where the biological and pathological influence has been reported. In silico analysis for rs241433 shows no sites of alternative splicing or miRNA generation. However, functional studies are required to establish the specific biological participation of this SNP.

Tapasin is encoded by the *TAPBP* gene. This protein plays an essential function in the peptides assembly previously translocated by TAP complex to HLA class I into the ER of cells [17]. It has been reported that some viral products compromise the antigen presentation by interacting with the Tapasin protein [41]. For example, the molluscum contagiosum virus encodes MC80 protein, which generates ER and Tapasin degradation, accompanied by a TAP loss, limiting HLA class I access to cytosolic peptides and a deficient antigen presentation [42]. Moreover, in a Chinese population, the rs9277972 was associated with an increased risk for hepatitis C virus [43]. On the other hand, the rs2071888 (SNP-associated in our study) has only been linked with gastrointestinal stromal tumors [44] and aspirin-exacerbated respiratory disease [45].

There are no previous studies where rs2071888 has been associated with influenza A/H1N1 or another viral infection. The rs2071888 is a missense variant in the *TAPBP* exon 4, which generates a Threonine > Arginine change, which could alter the polarity of the protein structure, thereby modulating interactions with other molecules such as the HLA class I, leading to an altered antigen presentation and consequently, deficient infected cells clearance by CD8+ T cells, higher viral replication, and disease development [45]. In this sense, we conducted a bioinformatics analysis where we found that the rs2071888/G allele decreases the affinity for a transcriptional factor, affecting the TAPBP expression, probably leading to a deficient antigen presentation. In addition, this SNP modifies the secondary structure of the mRNA, which could alter the protein function. Interestingly, in the in silico analysis, we corroborate that—as we described before—the rs2071888/G allele change from medium size and polar amino acid (threonine) change to large size and basis (arginine), which modifies the polarity of the protein structure [46,47].

We did not find significant associations with the three SNPs associated with influenza A/H1N1 infection in the mortality association analysis, possibly due to the reduced sample size. Other studies have evaluated different SNPs with disease severity (where mortality is included as a severe characteristic). García-Ramírez et al. [11] described the rs361525 in *TNF* associated with greater disease severity in the same Mexican population. In addition, Chen and colleagues [48] found that the IFITM3 rs12252 SNP is associated with an increased risk of severe and mild influenza in Asian and Caucasian populations.

Furthermore, in the haplotype analysis, we found an increased frequency of the GG haplotype in the INF-P group, shaped by the two common alleles of rs3763365 and rs9276810 (both in the *PSMB8* gene), as well as a higher frequency of the AG haplotype, shaped by minor alleles of rs3763365 (*PSMB8*) and rs4148882 (*TAP1*) in HC group. Previously, it was established that catalytic proteasome subunits—PSMB8 and PSMB9—were closely linked to the *TAP1* and *TAP2* genes, suggesting a critical role in processing and antigen presentation [41,49]. To the best of our knowledge, to date, no studies have determined whether the products of different PSMB alleles differ concerning the peptides produced. A previous study demonstrated that an SNP (rs2071543) in *PSMB8* is associated with a higher risk for HP [21]. Additionally, a study where rs2071543 was evaluated in cervical cancer showed that a common allele is associated with a higher risk for developing the oncological disease [38], which is in the same sense as our haplotype results, suggesting an essential role of the proteasome in viral clearance and antigen presentation.

Other studies have evaluated single-nucleotide variants in diverse genes. For example, Zúñiga et al. [50] found four polymorphisms associated with severe pneumonia in Influenza A/H1N1, including SNPs in the Fc fragment of immunoglobulin (Ig) G; low-affinity IIA; receptor (*FCGR2A*), RPA interacting protein (*RPAIN*); and complement component 1, q subcomponent binding protein (*C1QBP*) genes. In this sense, Chatzopoulou and colleagues [51] also found positive associations with SNPs in complement-associated genes (*FCGR2A*, *C1QBP*, *CD55*).

Our study is not free of limitations. Firstly, we included a relatively small sample size for both cases and control; however, our study was done with samples from our laboratory’s biobank and exclusively with samples from patients with a confirmed influenza A/H1N1 diagnosis. Secondly, we carried out a mortality analysis; however, we have a reduced sample size. Thirdly, we did not measure the expression for any of the genes evaluated. Despite the associated SNPs not meet HWE, we conducted bioinformatics and in silico analysis, where we found data supporting our genetic-association study.

## 5. Conclusions

In summary, patients with influenza A/H1N1 infection are predominantly men, obese, and smokers. The rs241433 AC genotype from *TAP2* and the AG haplotype (rs3763365 and rs4148882) are associated with a reduced risk for influenza A/H1N1 infection. The GG genotype from *TAPBP* rs2071888 and the GG haplotype (rs3763365 and rs9276810) are associated with a higher risk for developing influenza A/H1N1 infection in a Mexican mestizo population. On the other hand, these SNPs are not associated with a higher risk of mortality. More association and functional studies and replication in other populations are required to establish these SNPs’ participation in the influenza A/H1N1 infection pathogenesis.

## Figures and Tables

**Figure 1 viruses-12-01224-f001:**
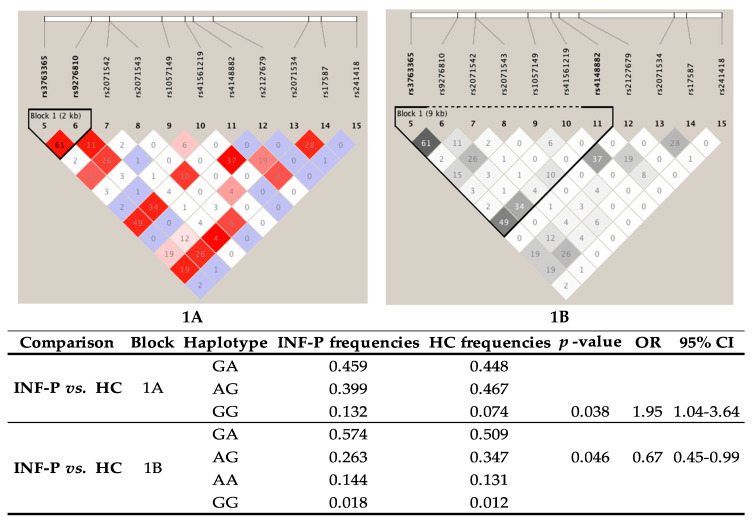
Haplotype analysis. (**A**) Haplotype conformed by the common allele of rs3763365 and rs9276810; (**B**) Haplotype conformed by minor alleles of rsr3763365 and rs41488882. INF-P—Patients with influenza A/H1N1 infection; HC—Healthy contacts. *p*-value < 0.05 was considered as significative.

**Table 1 viruses-12-01224-t001:** Demographic and clinical data of the influenza A/H1N1 infection (INF-P) and healthy contact (HC) groups.

Variables	INF-P = 128	HC = 111	*p*-Value
Age	41 (19–86)	44 (19–82)	0.986
Sex, male (%)	77 (60.16)	46 (41.44)	0.003
BMI	30.5 (17.4–52.8)	26.8 (16–44.8)	<0.001
Tobacco smoking, %	57 (44.53)	39 (35.14)	0.139

All values are expressed as median and minimum–maximum values. We used the Mann–Whitney *U* test and Fisher Exact test. *p*-value < 0.05 was considered as significant.

**Table 2 viruses-12-01224-t002:** Allele and genotype frequencies between cases and controls.

*Gene*	INF-P		HC		*p*-Value	*p*-Value Corrected	OR	95% CI
	n	F (%)	n	F (%)				
***TAP2***	**rs241433**							
Genotypes								
AA	75	58.59	28	25.23	<0.0001	<0.0001	4.19	2.41–7.30
AC	53	41.41	83	74.77			0.24	0.14–0.42
CC	0	0	0	0				
	128	100	111	100				
Alleles								
A	203	79	139	62.61	<0.0001	<0.0001	2.29	1.52–3.44
C	53	21	83	37.39			0.44	0.29–0.66
***TAPBP***	**rs2071888**							
Genotypes								
CC	24	18.90	28	25.45	0.0008	0.014	1	
CG	74	58.27	76	69.09			1.14	0.60–2.14
GG	29	22.83	6	5.45			5.64	2.00–15.86
	127	100	110	100				
Alleles								
C	122	48.03	132	60	0.009	0.153	0.62	0.43–0.89
G	132	51.97	88	40			1.62	1.13–2.34
***TAPBP***	**rs2282851**							
Genotypes								
CC	36	29.03	15	13.64	0.017	0.289	1	
CT	69	55.65	75	68.18			0.38	0.19–0.76
TT	19	15.32	20	18.18			0.40	0.17–0.94
	124	100	110	100				
Alleles								
C	141	57	105	47.73	0.048	0.816	1.44	1.00–2.08
T	107	43	115	52.27			0.69	0.40–0.99

INF-P—Patients with influenza A/H1N1 infection; HC—Healthy contacts. *p*-value < 0.05 was considered as significant.

**Table 3 viruses-12-01224-t003:** Logistic regression analysis for single nucleotide polymorphisms (SNPs) associated with influenza A virus (IAV) infection.

Alleles							
Chr	Gene	SNP	A1	Test	*p*-Value	OR	95% CI
6	*TAP2*	rs241433	C	Add	<0.0001	0.241	0.133–0.434
				Sex	0.03979	1.828	1.029–3.247
				BMI	<0.0001	1.102	1.05–1.157
	*TAPBP*	rs2071888	G	Add	0.0095	1.891	1.168–3.061
				Sex	0.0056	2.184	1.257–3.794
				BMI	0.0001	1.097	1.046–1.15
**Genotypes**							
**Chr**	**Gene**	**SNP**	**A1**	**Test**	***p*-Value**	**OR**	**95% CI**
6	*TAP2*	rs241433	CC	Add	NA	NA	NA
				Domdev	NA	NA	NA
				Sex	NA	NA	NA
				BMI	NA	NA	NA
				Geno_2DF	NA	NA	NA
	*TAPBP*	rs2071888	GG	Add	0.005	2.184	1.274–3.745
				Domdev	0.075	0.5612	0.297–1.059
				Sex	0.009	2.102	1.205–3.668
				BMI	0.0002	1.093	1.043–1.145
				Geno_2DF	0.013	NA	NA

Chr—Chromosome; SNP—Single Nucleotide Polymorphism; BP—Base pair location; A1—minor allele for each SNP; OR—Odds ratio; 95% CI—Confidence interval; Add—Additive effect model; Domdev—Dominance deviation; BMI—Body mass index; Geno_2DF—Genotypic 2 degree-freedom. *p*-value < 0.05 was considered as significant.

**Table 4 viruses-12-01224-t004:** Sequences and affinity for different proteins.

SNP Allele	Sequence	Absolute Affinity Value	Protein
rs241433/C	GGATATACAGTCCCTTCTCCTACCATACAG	12.4	UP1
rs241433/A	GGATATACAGTCCATTCTCCTACCATACAG	0.4	
rs2071888/G	TTGAACTGTAGGCAGCC	0.8	HNF4
rs2071888/C	TTGAACTCTAGGCAGCC	10.1	

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
