# Peer review of "Polymorphisms in Processing and Antigen Presentation-Related Genes and Their Association with Host Susceptibility to Influenza A/H1N1 2009 Pandemic in a Mexican Mestizo Population"

_viruses, 2020, doi:10.3390/v12111224_

Round 1
Reviewer 1 Report
The authors have addressed concerns. One minor recommendation is to state the population under study earlier in the abstract and if possible also in the title.
Author Response
Thank you very much. We have clarified that the study was carried out in a Mexican mestizo population in the title and abstract.
Reviewer 2 Report
The manuscript entitled "Polymorphisms in processing and antigen 2 presentation-related genes and their association with host susceptibility to influenza A (H1N1) 2009 pandemic” describes findings from clinical samples obtained from influenza patients during 2009-2012 influenza pandemic. Ponce-Gallegos and colleagues investigated the presence of 17 SNPs in their patient collectives (n=239). Herein, they found that rs241433/C allele, as well as AG haplotype (rs3763365 and 22 rs4148882), are associated with reduced risk for influenza A/H1N1 infection, while the rs2071888G allele (TAPBP) and GG haplotype (rs3763365 and rs9276810) are associated with a higher risk. In general, this descriptive study is well conducted and well presented.
- The description of the patient collective is poor. Is there any difference in the presence of risk factors between the two groups investigated? There is information in S2 table only for INF-P group.
- It would be interesting to see whether there is a difference in ORs between mild and severe patients in INF-P group. Have the authors examined these data?
- Is there a difference in susceptibility to influenza virus when adjusting for underlying comorbidities except BMI? These findings should be discussed this, so the reader can judge the relevance of the findings.
- It is confusing whether the SNPs studied are in HWE or not. In lines 186-190 and 254-256 the authors claim that out of 17 polymorphisms only 6 did not meet the HWE. The three main polymorphisms presented, namely rs241433, rs2071888 and rs2282851 is supposed to fulfill the equilibrium criteria. However, in line 282 the authors state that “Nevertheless, both SNPs did not meet the HWE”. Could you please clarify it?
- In Table 2 the data of OR and 95% CI for rs2282851 genotypes are missing.
- In the Conclusions section, lines 350-351, authors claim that “rs241441 AC genotype… associated with a reduced risk for influenza A/H1N1 infection”. This finding is not supported in the study.
- It would be nice to discuss findings in similar studies (for example Zuniga et al 2012, Chatzopoulou et al, 2019, Lee et al 2017) in the later years, since the literature is not such extensive in this field despite its high importance. The finding of such potential genetic polymorphisms predisposing for host susceptibility to influenza virus infections could be a very useful tool.
- Typing errors:
Line 216: “rs241433/C” instead of “rs241433/G”
Lines 224-226: p-value significance is missing
Line 257: Authors describe Figure 1A but in the figure there is not part A, B or C.
There are a few other grammatic or typing errors that should be corrected mainly in the Discussion section.
Author Response
The description of the patient collective is poor. Is there any difference in the presence of risk factors between the two groups investigated? There is information in S2 table only for INF-P group.
R= Thank you for your observation. Due to the control subjects were asymptomatic, we applied only a small questionnaire which did not include those variables.
It would be interesting to see whether there is a difference in ORs between mild and severe patients in INF-P group. Have the authors examined these data?
R= This is an interesting observation. Due to the “severe” disease criteria can be heterogeneous, we decided to carry out a mortality analysis to compensate for this limitation. We included a depicting table with demographic/clinical comparison between survivors and non-survivors (during hospitalization) and allele and genotype frequencies comparison. After that, we performed a logistic regression analysis adjusting for possible confounding variables. We did not find significant associations (rs241433/TAP2: p= NA; rs2071888/TAPBP: p= 0.25; rs2282851/TAPBP: p= 0.75). In the current version has been included this analysis.
Is there a difference in susceptibility to influenza virus when adjusting for underlying comorbidities except BMI? These findings should be discussed this, so the reader can judge the relevance of the findings.
R= Thank you. We did not include subjects with comorbidities in the control group, and then we did not perform a comparison between both groups to establish a significant difference with these variables. So, in the logistic regression analysis, we considered only variables that we included in both groups.
It is confusing whether the SNPs studied are in HWE or not. In lines 186-190 and 254-256 the authors claim that out of 17 polymorphisms only 6 did not meet the HWE. The three main polymorphisms presented, namely rs241433, rs2071888 and rs2282851 is supposed to fulfill the equilibrium criteria. However, in line 282 the authors state that “Nevertheless, both SNPs did not meet the HWE”. Could you please clarify it?
R= Thank you for this observation. As you said, 6 of the 17 evaluated SNPs did not meet the HWE, being the principal 3 SNPs in our study included in these 6 SNPs. We decided to continue with the study since in diverse publications is mentioned that mestizo populations (as Mexican) have a rich genetic variability, a product of the years of genetic recombination between ancestral populations (Amerindian, Caucasian, African and Asian descendants), which could result in some SNPs having a different behavior, as we described in the discussion section. Also, in the Limitations (at the end of discussion), we described that to strengthen our results, we carried out an In Silico and a Bioinformatic analysis with the associated SNPs with interesting results.
In Table 2 the data of OR and 95% CI for rs2282851 genotypes are missing.
R= Thank you for your observation. Now we have included the missing data.
In the Conclusions section, lines 350-351, authors claim that “rs241441 AC genotype… associated with a reduced risk for influenza A/H1N1 infection”. This finding is not supported in the study.
R= We are sorry for the confusion. We referred to the rs241433/AC SNP (TAP2 gene). Now we have corrected the mistake.
It would be nice to discuss findings in similar studies (for example Zuniga et al 2012, Chatzopoulou et al, 2019, Lee et al 2017) in the later years, since the literature is not such extensive in this field despite its high importance. The finding of such potential genetic polymorphisms predisposing for host susceptibility to influenza virus infections could be a very useful tool.
R= Thank you for your recommendation. Now we have included these studies in the discussion section.
Typing errors:
Line 216: “rs241433/C” instead of “rs241433/G”
Lines 224-226: p-value significance is missing
Line 257: Authors describe Figure 1A but in the figure there is not part A, B or C.
R= We apologies for the mistakes. Now we have made the corresponding changes.
There are a few other grammatic or typing errors that should be corrected mainly in the Discussion section.
R= Thank you for this observation. Now we have corrected the typos in the manuscript.
This manuscript is a resubmission of an earlier submission. The following is a list of the peer review reports and author responses from that submission.
Round 1
Reviewer 1 Report
In the manuscript by Ponce-Gallegos et al, the role of single nucleotide polymorphisms in several genes involved in MHC-1 antigen processing are evaluated for association with severe A/H1N1/pdm09 virus infection in a Mexican mestizo population. Two SNPs, in Tap2 as well as the TapBp were identified that either reduce or increase the risk of infection warranting hospitalization.
Minor and Major concerns are as follows:
- English grammar editing in a number of sentences is required.
- The two major SNPs of interest of the 17 evaluated that showed significant differences did not meet the Hardy-Weinberg Equilibrium quality control in the Mestizo population and may explain why this association has not been previously reported. The authors should validate whether the Tap and TapBp SNP associations hold within a population cohort that meets this quality control standard.
- Validation that the SNPs detected impact the presentation of viral antigens on HLA and ultimate CD8 T cell recognition and effector functions would increase the impact of the study.
Reviewer 2 Report
The manuscript identified several SNPs in processing and antigen presentation-related genes potentially linked to host susceptibility to influenza A (H1N1) 2009 pandemic virus. Although the study was limited by the small number of subjects, it nevertheless provides a clue on how the SNPs in key genes regulate disease outcome. Two identified SNPs including rs241433/C and rs2071888G allele were shown for the first time to be associated with reduced or increased risk to influenza virus infection respectively and their role in viral infection warrants further studies in the future.